



# Effective Storm Surge Evacuation Planning Coupling Risk Assessment and DRL: A Case Study of Daya Bay Petrochemical Industrial Zone

Chuanfeng Liu[2], Yan Li[1], Wenjuan Li[3], Hao Qin[2], Lin Mu[1,4], Si Wang[1,4], Darong Liu[2], and Kai Zhou[3]

[1]College of Life Science and Oceanography, Shenzhen University
[2]College of Marine Science and Technology, China University of Geosciences
[3]Shenzhen Marine Development and Promotion Center
[4]College of Computer Science and Software Engineering, Shenzhen University

**Correspondence:** Yan Li (liyan_ocean@szu.edu.cn), Wenjuan Li (lwenjuan@hotmail.com)

**Abstract.** Storm surge is one of the most destructive marine disasters, characterized by abnormal and temporary rises in water levels during intense storms, leading to extreme inland flooding in the coastal area. Emergency evacuation planning, based on storm surge risk assessments, plays a crucial role in saving lives and mitigating disasters. Conventional emergency evacuation plans primarily adopt the perspective of administrators, providing evacuees with complete environmental information. However, in practical situations, evacuees often lack access to complete environmental information and need to select appropriate paths based on their limited awareness of their surroundings. This study coupled a risk assessment of storm surges with a road network to optimize evacuation routes in the Daya Bay Petrochemical Industrial Zone, a low-lying coastal region of Huizhou City, China, which is frequently affected by storm surge-driven flooding. A combination of the Deep Q-Network (DQN) model and raster environment was employed to develop real-time evacuation plans based on limited surrounding environments during storm surge events. To address the DQN model's convergence challenges, masked state space, masked action space, and tri-aspect reward mechanism were proposed, profoundly enhancing the model's convergence capabilities. The coupled ADCIRC-SWAN model and the Jelesnianski hurricane model were utilized to simulate storm surges for risk assessments under various typhoon scenarios. Additionally, potential safe shelters were identified to offer alternative evacuation options. Two distinct storm surge scenarios were employed as test environments, evaluating path plans for 1000 randomly selected starting points in each case. The results indicate that the proposed method is highly effective in devising optimal evacuation routes with minimal deviation, offering valuable guidance for evacuees during real-world storm surges.

## 1 Introduction

A storm surge is an abnormal and temporary rise of water that occurs during intense storms. This sudden rise in sea level can lead to extreme inland flooding in coastal communities especially when an advancing surge coincides with astronomical high



tide (Wang et al., 2021b). Storm surge is one of the most dangerous and destructive natural hazards to life and property along the coastline and kilometers inland in the world (UNISDR et al., 2015). In the US, coastal flooding from storm surge was responsible for 49 % of hurricane- or tropical-storm-related fatalities during the period from 1994 to 2003 (Rappaport, 2014). When Hurricane Katrina struck the southeastern United States in 2005, an estimated 1,577 people died, causing USD 108

billion in property damages (Rhome and Brown, 2006). The storm surge from Super Typhoon Haiyan, hitting the Philippines in 2013, was estimated to be 7.5 m high, leading to more than 7,000 persons losing their lives along the coastline (Mas et al., 2015). In China, from 1998 to 2019, the average annual economic losses resulting from storm surge flooding each year is approximately 10.17 billion RMB, which is equivalent to 96 % of the total direct economic losses from all types of marine disasters (China Marine disaster bulletin, 2021). Recent studies suggested that the number of people facing the risk of storm

surge flooding and the losses in property and human lives during typhoon events could continue to increase in the future due to accelerating sea-level rise and more intense hurricanes (Merkens et al., 2016; Oppenheimer et al., 2019; Snaiki et al., 2020). Physical barriers, which run parallel to the shoreline, alone cannot prevent all possible damages that urban settlements and infrastructure can suffer during storm surge flooding. With increasing potential victims and economic losses, it is of paramount importance to perform risk assessments and develop evacuation plans to mitigate the risks associated with storm surges.

Storm surge risk assessment entails the identification and evaluation of potential hazards and associated risks within a specified region, as well as the severity of their potential consequences (Wang et al., 2021a). Conducting effective and practical storm surge risk assessments lays the groundwork and establishes the premise for devising well-founded evacuation plans. Generally, the inundation depths and extents are regarded as the most common factors to measure the tangible risk, and the computation of potential storm surge heights and the maximum inundated area under different hypothetical(Thieken et al.,

2007; Merz et al., 2010) typhoon events is one of the major components of storm surge risk assessment. The computation of maximum potential impact due to storm surge is usually performed using the numerical model by taking into account the atmospheric pressure, landfall location, varying forward speed, the radius of maximum wind, and tracking. So far, many hydrological models such as the Advanced Circulation (ADCIRC, Luettich et al.) model, Finite Volume Community Ocean Model (FVCOM, Chen et al.), and Simulating Waves Nearshore (SWAN, Delft University of Technology) model have been

applied to simulate tide, wave and storm surge in different regions in the world, obtaining good prediction accuracy in previous studies. In this study, the coupled ADCIRC+SWAN model was employed to simulate storm surge flooding, and quantitative risk assessments were conducted by combining the extents and depths of the flooding.

The research on emergency evacuation originated in the early 20th century, and its core task is the development of evacuation plans, including the identification of disaster shelters and the planning of evacuation routes (Alsnih and Stopher, 2004). In

some coastal areas vulnerable to storm surges, there might be a lack of designated disaster shelters. Consequently, it is essential to identify suitable facilities to serve as shelters, and a framework was proposed based on several criteria, including structural stability, waterproofing, and capacity. Conventional shortest-path algorithms, such as Dijkstra's and A* algorithms, have been enhanced and employed in emergency evacuation planning (Samah et al., 2015; Astri et al., 2020, e.g.,). Additionally, heuristic approaches, including the PSO algorithm, genetic algorithm, and ant colony algorithm have been introduced to identify optimal

routes within intricate environments (Li et al., 2020; Goerigk et al., 2014; Forcael et al., 2014, e.g.,). Nevertheless, existing





emergency evacuation plans predominantly adopts the perspective of administrators, furnishing evacuees with complete environmental information. In realistic circumstances, evacuees often encounter obstacles in obtaining the entire environmental information, instead possessing merely a limited awareness of their immediate surroundings. And the techniques employed for issuing evacuation mandates and cautionary advisories lack the timeliness in guiding individuals to the most appropriate escape
routes where actual evacuation distances may fluctuate as the evolving environment of the catastrophe. Deep reinforcement learning (DRL) is a new paradigm of deep learning, and since its inception, it has rapidly evolved with the proposal of various algorithms, such as Deep Q-Network (DQN, Mnih et al.), Asynchronous Advantage Actor-Critic (A3C, Mnih et al.), and Proximal Policy Optimization (PPO, Schulman et al.). DRL has found widespread application in diverse domains, including game-playing (Mnih et al., 2015), autonomous navigation (Sallab et al., 2017), and industrial regulation (Levine et al., 2018),
demonstrating strong learning and generalization capabilities. The merit of DRL algorithms in emergency evacuation stems from their capacity to function without a priori comprehension of the entire environment. In this study, an enhanced deep reinforcement learning method was proposed to assist individuals in evacuating to the nearest shelter based on their surrounding environment. Additionally, to tackle convergence challenges in path planning using DRL in large-scale areas, two compression methods were proposed to significantly reduce the problem size.

The case study area of this research focuses on the low-elevation coastal regions of the Daya Bay district, which is periodically exposed to tropical cyclones and frequently affected by storm surge-driven coastal flooding. The work in this study involves: 1. employing the coupled ADCIRC+SWAN model to simulate storm surge flooding for risk assessments; 2. identifying disaster shelters; 3. constructing a simulation environment; 4. utilizing enhanced DQN for evacuation path planning. And the main contribution is to develop a real-time effective emergency evacuation plan for individuals with limited awareness of
their surrounding environment in a large-scale region. The rest of the paper is organized as follows: Section 3 delineates the coupled numerical model utilized to simulate storm surge flooding during typhoon events and the deep reinforcement learning algorithm used to explore optimal routes to evacuate from the floods on the road network. Thereafter, in the Section 4, the implementation of the proposed method to simulate depths and extents of storm surge flooding and recommended evacuation routes is demonstrated with the example of the coastal area of Huizhou. Subsequently, the findings and analyses pertaining to
the numerical simulations of inundation depths and extents of storm surge flooding, as well as the emergency flood evacuation simulations based on DRL in coastal regions, are presented. Finally, the conclusions and suggestions for future work were discussed in Section 5.

## 2  Study Area

Daya Bay district is located in the southern region of Huizhou City, Guangdong Province. It has a total land area of 293 $\text{km}^2$
and a population of 0.45 million, which is concentrated most highly in coastal areas, in 2021. In addition, the Daya Bay Petrochemical Industrial Zone, situated in the north-eastern part of Daya Bay, was listed as a national petrochemical industrial base. It has formed an annual production capacity of 22 million tons of oil refining and 2.2 million tons of ethylene in 2021, which ranks first in China in terms of the scale of petrochemical-refining integration. Industrial facilities and critical infrastructure



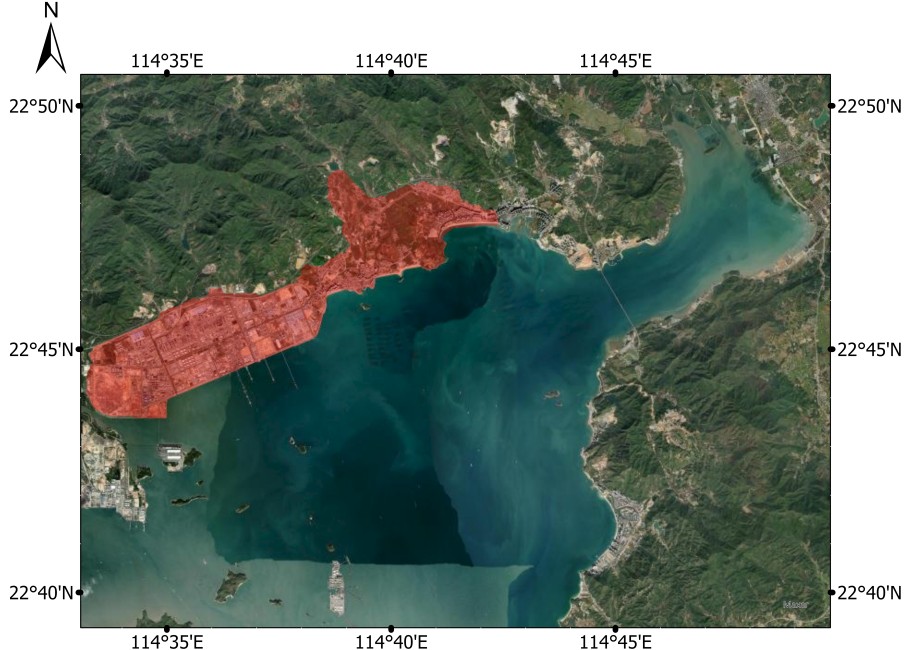

**Figure 1.** The Daya Bay. The study area is in red.

in this area are vulnerable to storm surge-driven coastal flooding during typhoon events, leading to devastating losses of life
and property. The peak water level induced by severe storm surge during super typhoon Mangkhut on 16 September 2018 at
Huizhou gauging station, closest to the Daya Bay district, rose to 349 cm, causing estimated economic losses of 577.39 million
RMB. In the context of substantial sea-level rise and urban extent along low-lying coastal areas, most communities across the
Daya Bay district will likely face higher storm surge flooding risk in the future. It is crucial and essential to create the storm
surge risk maps for raising awareness about areas at risk and making evacuation plans to minimize the loss and damage. The
study area is shown in Fig. 1.

## 3  Methodology

As storm surges assail, the goal of storm surge evacuation planning is to provide guiding paths to shelters for the people in
the affected area, despite their limited environmental knowledge. The significance of this work lies in the ability to simulate
storm surge flooding for risk assessments and to find routes to the nearest disaster based on the road network. In this study, an
effective method for planning the evacuation path during a storm surge is proposed. The specific research method includes two
segments: environmental modeling and deep reinforcement learning. In Section 3.1, the process of simulating storm surges
using numerical techniques and establishing an authentic raster environment model for the study area is delineated. In Section



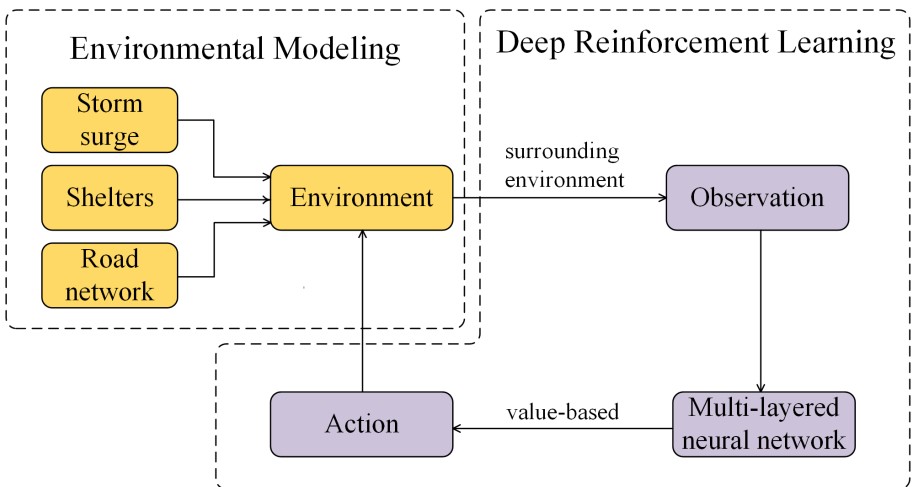

**Figure 2.** The overview of the work.

3.2, the principles and procedures of the Markov Decision Process (MDP) are described. Based on the MDP, we redefined the path planning problem with the raster environment and proposed two optimization methods to reduce the problem scale. Please refer to Fig. 2 for an overview and the sections for more information.

## 3.1  Environmental modeling

The purpose of environmental modeling lies in the accurate simulation of the environmental information of the study area. This work involves storm surge simulation and risk assessments, regional road network modeling, and identification of potential disaster shelter facilities. To facilitate computational engagement, all environmental data have been rasterized. The rasterized environment, with a three-dimensional construct, consists of regional risk levels, road networks, and disaster shelters, and these constituent elements are stored within three separate matrices. The whole flow chart of environmental modeling is depicted in Fig. 3.

### 3.1.1  Simulation of storm surge and risk assessments

Storm surge is a phenomenon wherein seawater levels experience a marked elevation due to turbulent atmospheric perturbations, such as typhoons and cyclones. The ADCIRC model, designed to address two- and three-dimensional hydrodynamic free surface circulation challenges, has been widely applied to simulate tide- and wind-driven circulations. The SWAN model is a numerical wave model, which computes the wave action density spectrum by solving the wave action balance equation. In this study, the Jelesnianski hurricane model (Jelesnianski, 1965) was utilized to generate wind field, and the coupled ADCIRC+SWAN model, which integrates storm surge and wave interactions, was employed to simulate typhoon-induced storm surge flooding during typhoon events. The computational domain in this study covered the coastal region of Huizhou, as





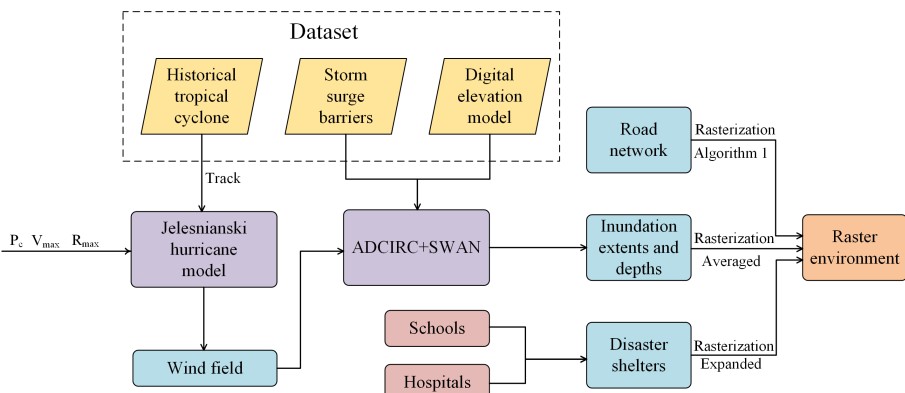

**Figure 3.** The flow chart of environmental modeling.

shown in Fig. 4. The model contained grids consisting of 74328 units and 38407 nodes. The coverage extended from 106.0° E to 128.0° E in longitude and from 13.0° N to 28.0° N in latitude. The 11 major tidal components of semidiurnal and diurnal frequencies (M2, N2, S2, K2, K1, O1, P1, Q1, MS4, M4, M6) were included. The Jelesnianski hurricane model calculates the air pressure and the wind. The ADCIRC model transfer wind field, water levels, and currents to the SWAN model every 600 s, while the SWAN model passes the wave radiation back to the ADCIRC model on the same unstructured finite element mesh, which can simulate storm surge and produce coastal flooding for the study area.

The coupled model is driven by the wind field, rendering the accuracy of storm surge model outcomes intrinsically reliant on the quality of the wind field model. Given the challenges in obtaining measured data, a prevalent approach involves constructing a theoretical wind field model based on the wind gradient formula (Jelesnianski, 1965; Willoughby and Rahn, 2004). Utilizing the Jelesnianski model necessitates four parameters: the cyclone track $T$, the minimum central pressure $P_c$, the maximum wind velocity $V_{max}$, and the maximum wind radius $R_{max}$. The $R_{max}$ may be approximated through several empirical equations (Vickery et al., 2000; Cheung et al., 2007), as delineated below:

$$R_{max\_1} = \exp(2.635 - 0.00005086\Delta P^2 + 0.0394899\theta) \tag{1}$$

$$R_{max\_2} = 1119 \times \Delta P^{-0.806} \tag{2}$$

$$R_{max\_3} = R_k - 0.4 \times (P_c - 900) + 0.01 \times (P_c - 900)^2 \tag{3}$$

$$R_{max\_4} = 260.93 \times \Delta P^{-0.512} \tag{4}$$

where $\Delta P$ indicates the pressure difference between the minimum central pressure $P_c$ and ambient pressure, and in this study, the ambient pressure is $1010 hPa$; $\theta$ represents the latitude of the storm's center; $R_k$ is an empirical constant usually taking the value range of [30,60], and we take the $R_k = 50$. The $\{R_{max\_1}, R_{max\_2}, R_{max\_3}, R_{max\_4}\}$ are all estimates of $R_{max}$ and the final $R_{max}$ we adopted was their average, i.e.

$$R_{max} = \frac{R_{max\_1} + R_{max\_2} + R_{max\_3} + R_{max\_4}}{4} \tag{5}$$



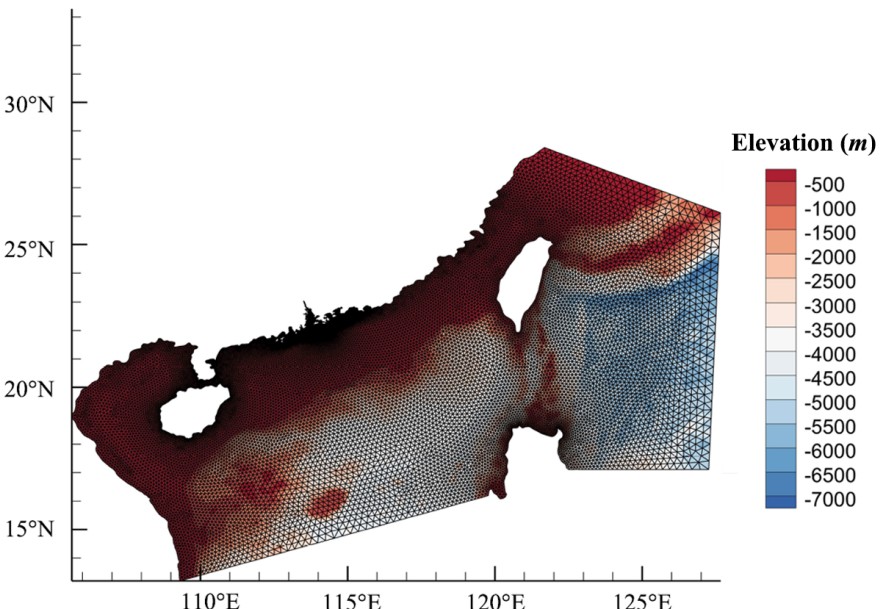

**Figure 4.** The computational domain.

Given a pressure difference $\Delta P$, the maximum wind velocity $V_{max}$ can be estimated by a empirical equation(Atkinson and Holliday, 1977):

$$V_{max} = 3.7237 \times \Delta P^{0.6065} \tag{6}$$

To simulate the inundation extents and depths under a specific typhoon, both the Digital Elevation Model (DEM) dataset and the Storm Surge Barriers (SSB) dataset were employed. The DEM dataset, available from Huizhou Land and Resource Bureau, is a raster dataset containing the land elevation in Huizhou. The SSB dataset, available from Huizhou Oceanic Administration, contains information on storm surge barriers, such as location, height, and slope. Storm surge risk assessment was conducted by evaluating the difficulties of traversing areas with varying inundation depths (Guidelines for Risk Assessment and Zoning of Marine Hazards Part 1: Storm Surge, 2019), and areas are classified into five risk levels, as displayed in Table 1, where in

Risk I areas, passage is relatively unimpeded; Risk II areas present moderate obstacles to traversal; Risk III areas pose a certain level of danger, making them impassable for vulnerable individuals such as children and the elderly; Risk IV exhibit elevated risks, with traversal incurring considerable costs; finally, Risk V areas present extreme danger, rendering passage virtually impossible.



**Table 1.** Risk levels and inundation depths

| Risk level | Inundation depth (m) |
|------------|---------------------|
| I | 0.0∼0.15 |
| II | 0.15∼0.5 |
| III | 0.5∼1.0 |
| IV | 1.0∼2.0 |
| V | 2.0∼ ∞ |

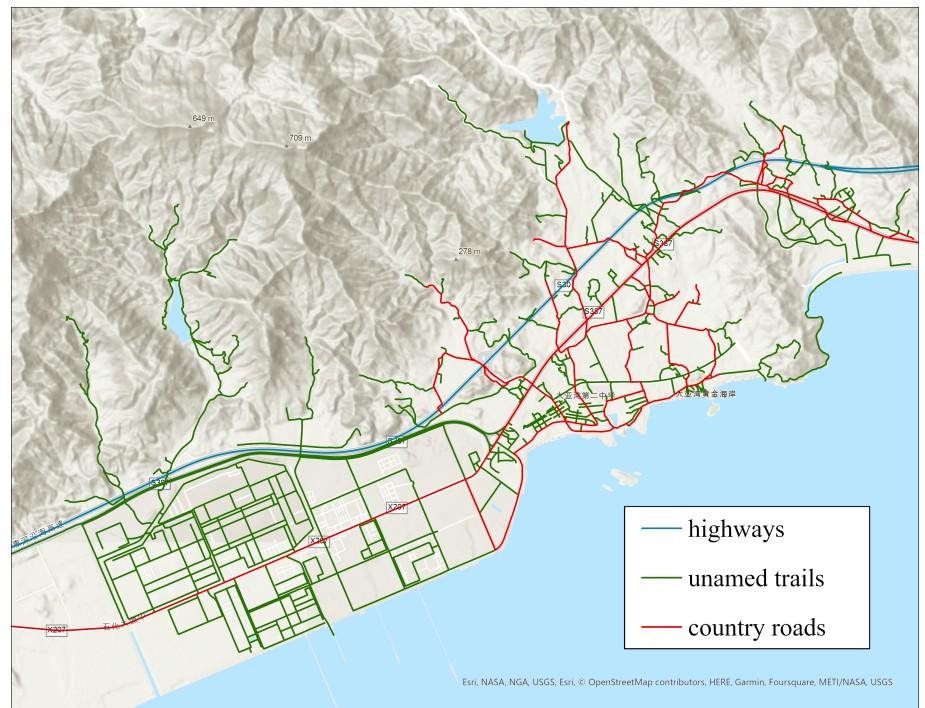

**Figure 5.** The road network in study area. The red line represents the country roads, the blue line represents the highways, and the green line represents the unnamed trails.

### 3.1.2 Rasterization of road network

In contrast to the orderly thoroughfares of urban areas, suburban roads tend to exhibit meandering and disjointed configurations. To enhance the road network within the study area, we collected additional unnamed trails by using Geographic Information System (GIS) techniques. These trials are branches and extensions of existing roads and highways. The road network in the study area is depicted in Fig. 5. Although the road network exhibits full connectivity, its complexity hampers the construction of a topological structure. To facilitate spatial analysis and deep reinforcement learning on the road network, vector road data





need to be rasterized. Raster data, akin to an image stored in a matrix, boasts a simplistic structure and readily engages in computational processes. In this study, an efficient method was proposed to convert road vector data into raster data. Road rasterization entails rasterizing each constituent vector segment. For a vector $v$ that starts at $(x_s, y_s)$ and ends at $(x_e, y_e)$, the rasterization algorithm is described in Algorithm 1, where $\{c_{i,j}\} = \{c_{0,0}, c_{1,0}, c_{1,1}, \ldots, c_{M,N}\}$ are the shorthand notations for

---

**Algorithm 1** Vector Rasterization

---

**Input:** $x_s, y_s, x_e, y_e, w, h$

**Initialize:** $\{c_{i,j}\} = 0, x_0 = \lfloor \frac{x_s}{w} \rfloor \cdot w + \frac{w}{2}, y_0 = \lfloor \frac{y_s}{h} \rfloor \cdot h + \frac{h}{2}, x_c = x_0, y_c = y_0, c_{x_0,y_0} = 1, \Delta x = x_e - x_s, \Delta y = y_e - y_s$

Solve the analytical formula for the vector $v$:

$v : \frac{y - y_s}{y_e - y_s} = \frac{x - x_s}{x_e - x_s}, x \in [x_s, x_e], y \in [y_s, y_e]$

Define the function $\mathcal{F}(a, b)$ as:

$\mathcal{F}(a, b) = \begin{cases} 1, a = b \\ 0, other \end{cases}$

**repeat**

    **if** $v$ intersects the line $x = x_c + (\frac{\Delta x}{|\Delta x|}) \cdot (\frac{w}{2})$ at $(x_p, y_p)$ **then**

        $x_c = x_c + \mathcal{F}(\max(|x_p - x_c|, |y_p - y_c|), \frac{w}{2}) \cdot w$

    **end if**

    **if** $v$ intersects the line $y = y_c + (\frac{\Delta y}{|\Delta y|}) \cdot (\frac{h}{2})$ at $(x_p, y_p)$ **then**

        $y_c = y_c + \mathcal{F}(\max(|x_p - x_c|, |y_p - y_c|), \frac{h}{2}) \cdot h$

    **end if**

    Determine the matrix coordinate of the cell:

    $i = \frac{x_c - \frac{w}{2}}{w}$

    $j = \frac{y_c - \frac{h}{2}}{h}$

    The vector passes through the cell $c(i, j)$:

    $c_{i,j} = 1$

**until** $|x_e - x_c| \leq \frac{w}{2}$ and $|y_e - y_c| \leq \frac{h}{2}$

**Output:** $\{c_{i,j}\}$

---

the set of cells; $(x_0, y_0)$ is the center coordinate of the starting cell; $\max(|x_p - x_c|, |y_p - y_c|)$ is the Chebyshev distance from $(x_p, y_p)$ to $(x_c, y_c)$. In this study, the Chebyshev distance is used to measure the distance between two cells in a raster setting. The procedure delineated in Algorithm 1 permits parallelization, enabling simultaneous rasterization of multiple vectors and thereby significantly accelerating the rasterization process for the entire road network. To distinctly differentiate individual road branches, the cell dimensions of w× h were established as 16 m×16 m. For a given area of size $H \times W$, the area can be rasterized into $M \times N$ cells and M=$\lceil W/w \rceil$, N=$\lceil H/h \rceil$, where $h \times w$ is the cell size. Each cell is independent, with the center





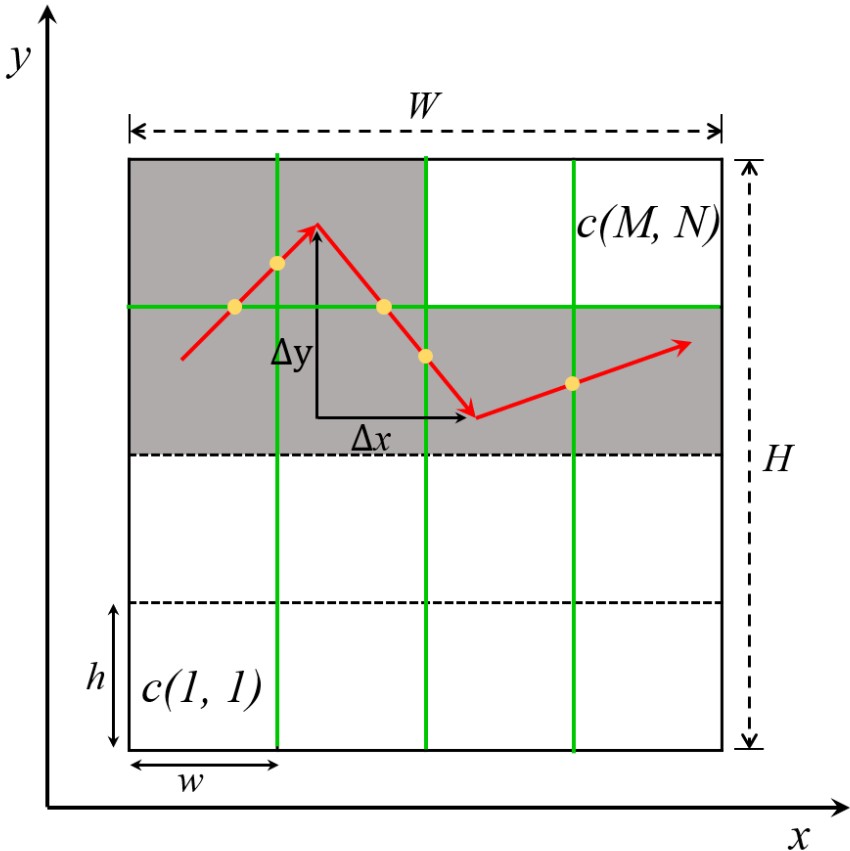

**Figure 6.** The rasterization of a road. A red straight line is a vector segment and it can be described using $\Delta x$ and $\Delta y$; The gray cells represent the result of the rasterization; The green lines are the boundaries of the adjacent cells that the vector traverse and the yellow points are the intersections.

coordinates of every cell representing the cell's location coordinates. A vector road is composed of a series of interconnected vector segments, arrayed sequentially from beginning to end, as illustrated in Fig. 6.

### 3.1.3 Identification of disaster shelters

Identifying potential safe shelters is crucial to enable swift evacuation of affected individuals to structurally sound and wa-
tertight facilities. Contrary to urban areas, dedicated disaster shelter facilities are scarce in suburbs, necessitating alternative options. This study employs three criteria—structural stability, waterproofing, and capacity—to identify two primary facility types as potential shelters. Hospitals are inherently designed to withstand various natural disasters. As per Chinese codes GB50011-2021 and GB50345-2021, hospital buildings are classified as Class II waterproof and Class I seismic-resistant struc-



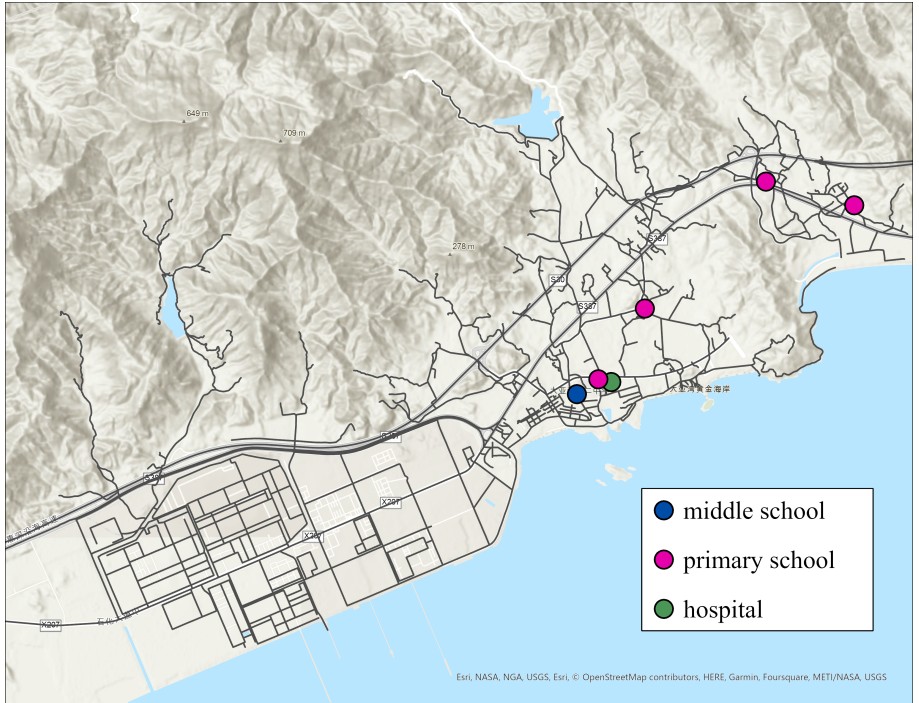

**Figure 7.** The spatial distribution of potential shelters in study area.

tures, featuring multiple waterproof amenities and robust earthquake resistance. Consequently, hospitals are prioritized as dis-

180 aster shelters during storm surges. In addition to hospitals, schools are designated as potential shelters. According to Chinese codes GB50011-2021 and GB50223-2021, school buildings are classified as Class III waterproof, and their seismic-resistant construction must surpass local residential buildings by one degree. Furthermore, hospitals and schools are typically situated in spacious, open areas, minimizing the likelihood of water accumulation. As public facilities, they can accommodate numerous evacuees. Fig. 7 depicts the spatial distribution of the chosen disaster avoidance facilities.

In this study, path planning relies on the road network, necessitating both the starting point and destination to be situated on roads. Given that shelter facilities (hospitals and schools) are generally not directly on roads, reaching road cells within a specified range from a shelter is deemed equivalent to reaching the shelter itself. This range is referred to as the 'shelter range'. With the road network cell size of 16 m×16 m, the 'shelter range' is defined as the Chebyshev distance of 8 cells (while the actual distance is 128 m) from the shelter. All road network cells within the "shelter range" are considered as destinations.

**3.2 Deep Reinforcement Learning**

Deep reinforcement learning is a new deep learning paradigm that focuses on formulating suitable policies and taking action to achieve a specific goal. A DRL agent learns autonomously through continuous interactions with a complex environment by



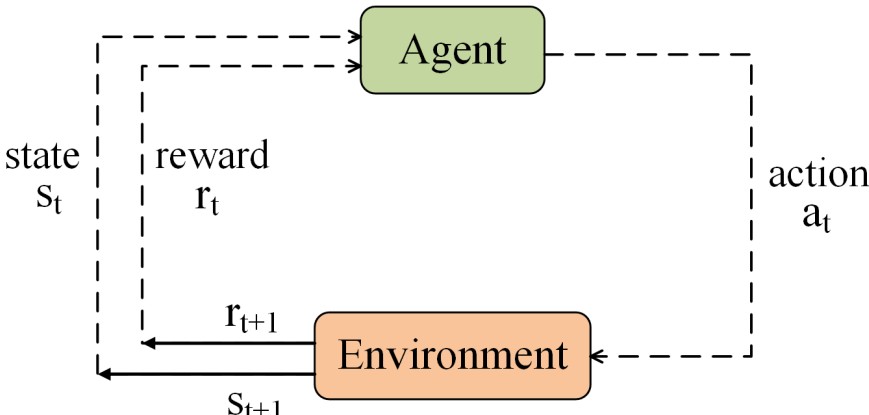

**Figure 8.** The DRL model. The environment is objective and refers to everything outside the agent that interacts with him, and the knowledge of the agent is reinforced through the interaction agent-environment. Consider the term state as the current environmental feature, i.e. all the information that the agent perceives from the environment at the current moment. In order to assess a policy specifically, the feedback mechanism called reward is introduced to define the value of a state and to quantify the effect of an agent's action.

performing actions and receiving rewards without explicit supervision. The interaction model between the agent and environment is illustrated in Fig. 8. The primary objective of the agent is to maximize cumulative rewards, which can be advantageous for evacuation planning. In this study, the Deep Q-Network was employed to maximize the cumulative rewards.

### 3.2.1 Markov Decision Process and Deep-Q Network

In the one-order Markov chain, the probability of a state $s_t$ is only related to the preceding state $s_{t-1}$, and such a property is called the Markov property. Based on the one-order Markov chain, given a set of states $S$, a set of actions $A$, and a set of rewards $R$, the Markov decision process (MDP, Bellman) and the Markov reward process (MRP, Bertsekas) are defined as Eq. 7 and Eq. 8:

$$P_{s,s'}^a = P(s_{t+1} = s' | s_t = s, a_t = a) \quad (s, s' \in S, a \in A) \tag{7}$$

$$R_{s,s'}^a = R(r_{t+1} | s_t = s, a_t = a, s_{t+1} = s') \quad (s, s' \in S, a \in A, r \in R) \tag{8}$$

where $P_{s,s'}^a$ represents the transition probablility from state $s$ to state $s'$ after performing action $a$, and $R_{s,s'}^a$ is the reward obtained after transition $(s, a, s')$. MDP can be regarded as a continuous decision-making process, and the next action to be performed is only dependent on the current state.

Value-based DRL algorithms try to estimate the value of states and actions. State value is the expected reward that the agent can obtain from a state to give an estimate of how good a state is, and action value is the expected reward that the agent can obtain from a state after performing a specific action, providing an estimate of the action's utility. under a policy $\pi$, the value





of a state $s$ is denoted as $V_\pi(s)$ and the value of a action $a$ in the state $s$ is denoted as $Q_\pi(s,a)$:

$$V_\pi(s) = \mathbb{E}_\pi(G_t|s_t = s) = \sum_a P_\pi(a|s)Q_\pi(s,a) \tag{9}$$

$$Q_\pi(s,a) = \mathbb{E}_\pi(G_t|s_t = s, a_t = a) = \sum_{s'} P^a_{s,s'}[R^a_{s,s'} + \gamma V_\pi(s')] \tag{10}$$

where $G_t$ is the total discounted reward from state $s_t$ and $\gamma \in [0,1]$ is the discount factor.

$$G_t = R_{t+1} + \gamma R_{t+2} + \ldots = \sum_{k=0}^{\infty} \gamma^k R_{t+1+k} \tag{11}$$

The goal of DRL is to find an optimal policy to maximize the state value and action value. Under the optimal policy $\pi^*$, the optimal state value function $V_*(s)$ and the optimal action value function $Q_*(s,a)$ can be obtained by:

$$V_*(s) = \max_a \sum_{s'} P^a_{s,s'}[R^a_{s,s'} + \gamma V_*(s')] \tag{12}$$

$$Q_*(s,a) = \sum_{s'} P^a_{s,s'}[R^a_{s,s'} + \gamma \max_a Q_*(s',a')] \tag{13}$$

which are called the Bellman optimality equations (Bellman, 2010). Consequently, the optimal state value is the highest attainable discounted reward from the state.

The path planning problem can be transformed into a finite and continuous decision-making process, wherein the agent chooses the subsequent action based on the current state until it reaches the destination. The location of the agent serves as the state, and the state transition is memoryless, satisfying the Markov property. Under the fixed-size raster environment and the same-size cells setting, the basic state space is $S : \{c_1, c_2, \ldots, c_{M \times N} | c_i = (x_i, y_i), 0 \leqslant x_i \leqslant M, 0 \leqslant y_i \leqslant N\}$, and the basic action space is potential moves to the 8 adjacent cells $A : \{a_1, a_2, \ldots, a_8 | a_i = (x,y), (x,y) \in \{\{-1,0,1\}^2 - (0,0)\}\}$. while the following state transition equation is available:

$$c' = c + a \quad (c, c' \in S; a \in A) \tag{14}$$

The path planning problem in raster environmrnt based on the MDP is defined as follows:

$$P^a_{c,c'} = P(c_{t+1}|c_t = c, a_t = a) = P(c_t = c, a_t = a) = 1 \tag{15}$$

$$\sum_{a \in A} P^a_{c,c'} = 8 \tag{16}$$

$$R^a_{c,c'} = R(r_{t+1}|c_t = c, a_t = a, c_{t+1} = c') = R(r_{t+1}|c_t = c, a_t = a) = f(d(c,c_e) - d(c',c_e)) \tag{17}$$

where $d(c,c_e)$ is the Chebyshev distance between cell $c$ and the destination cell $c_e$. In the study area, the destination is represented by a cluster of shelter facilities, collectively constituting the destination set $D$. Let $d(c) = \min d(c, c_e), c_e \in D$ signifies the distance to the nearest shelter facility. We defined the reward $R^a_{c,c'}$ as a function of the difference $d(c, c_e) - d(c', c_e)$, implying that the reward is related to the agent's proximity to the destination. If $c'$ is closer to the destination, the reward is positive,



and vice versa. The corresponding state value function and action value function are:

$$V_\pi(c) = \sum_{a \in A} P_\pi(a|c) Q_\pi(c,a) \tag{18}$$

$$Q_\pi(c,a) = f(d(c,c_e) - d(c+a,c_e)) + \gamma V_\pi(c+a) \tag{19}$$

A DQN is a multi-layered neural network, capable of approximating the optimal action value function $Q_*(c,a)$. Essentially, this function maps the n-dimensional state space to the action space. DQN is a value-based DRL algorithm where the output

for a given state $c_t$ is a vector of action values denoted as $Q(c_t, \cdot; \theta)$, with $\theta$ representing the parameters of the online network. The agent's policy is to perform the action associated with the highest value. Moreover, DQN employs an experience replay mechanism (Mnih et al., 2015), where past experiences are stored in a memory buffer and randomly sampled for training, to break the temporal correlation between samples and enable the agent to learn from infrequent events. Another important feature of DQN is the use of a separate target network (Mnih et al., 2015) for estimating the $Q_*(c,a)$, thereby enhancing the stability

of the learning process. The DQN searches for the optimal policy to maximize the largest long-term cumulative reward that the target is:

$$Y_t^{\text{DQN}} \equiv r_{t+1} + \gamma \max_a Q\left(c_{t+1}, a_t; \boldsymbol{\theta}_t^-\right) \tag{20}$$

and using the Mean squared error loss (RMSE) as loss function, the DQN can be trained by optimizing the following loss:

$$Loss(\theta_t) = \mathbb{E}[(r_t + \gamma \max_{a_{t+1}} Q\left(c_{t+1}, a_{t+1}; \boldsymbol{\theta}_t^-\right) - Q(c_t, a_t; \theta_t))^2] \tag{21}$$

The update process is based on the Monte Carlo method. By continuously interacting with the environment, the agent observes immediate rewards and accumulates them to count value information, which can then be transformed into a regression problem.

### 3.2.2    Addressing Convergence Challenges in DQN

The Deep Q-Network (DQN) model, when applied in a raster environment with each cell possessing eight neighboring cells, faces significant convergence challenges from two aspects: 1. the extensive search space, and 2. the issue of sparse rewards.

The fundamental state space, consisting of over 19,000 road cells, results in a vast search space of around $8^{19000}$. Coupled with the issue of sparse rewards, where the agent receives feedback only occasionally, the model's training process becomes even more complex and the convergence becomes notably difficult.

To counteract the vastness of the search space, two innovative methods were proposed: the masked action space and the masked state space. These methods effectively reduce the search space, thereby aiding in the model's convergence. In the basic

action space, eight moves are available from the current cell in various directions. However, many of these actions might be meaningless. By leveraging a mask, the compressed action space method efficiently narrows down the available actions from 8 to an approximate average of 3 per cell. This method focuses particularly on the transition of the action space, leading to dynamic action spaces where the action space transition is intricately linked with the state transition. For each cell (state) $c$, there is an associated action space $A_c$, where $|A_c| \leqslant 8$, and the action space transition is dependent on the state transition, as



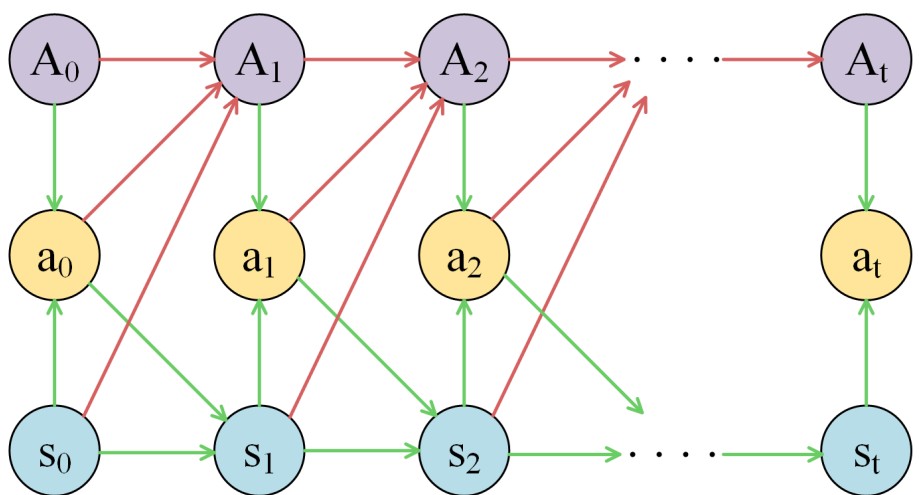

**Figure 9.** The action space transition.

illustrated in Fig. 9, where the red line represents the state transition and the green line represents the action space transition. According to Eq. 14, the action space of the current state relies on the preceding state and the last performed action, rendering the transition of the action space in compliance with the Markov property. The transition of action space is defined as:

$$P_{a,a'}^c = P(a_{t+1} = a'|c_t = c, a_t = a) = \pi(c+a); \tag{22}$$

where $a \in A_c$, $a' \in A_{c+a}$, $P_{a,a'}^c$ represents the probability of taking action $a'$ after taking action $a$ in cell $c$. There are two special cases in path planning on a raster environment: (1) For a state transition $c' = c + a$, an action $a'$ exists such that $c = c' + a'$, in this case the action $a'$ is deemed meaningless for state $s'$. (2) Moving one cell in the diagonal direction is essentially equivalent to moving one cell both horizontally and vertically. In this case, given the following transitions: $c + a = c'$, $c' + a' = c''$, and $c + a'' = c''$, the action $a'$ is meaningless after $c + a$. Fig. 10 depicts two of eight compressed action spaces, namely action patterns. As path planning operates on the extant known road network, exploring areas devoid of roads is considered futile. By focusing exclusively on state transitions within the road network, the action space can be further compressed. The compressed action space $A(c')$ for each state transition $c + a$ can be calculated using the raster road network and the eight action patterns.

$$A(c') = I \circ A\_P(a) \circ R\_N(c+a) \tag{23}$$

where $I$ is the basic action space of size $3 \times 3$ with all elements equal to 1, $R\_N$ is the matrix of the road network and $R\_N(c)$ is the mask of size $3 \times 3$ centered at cell $c$, the $A\_P(a)$ is the action pattern for $a$, and the operator $\circ$ is the Hadamard product. For each pair $(c, a)$, the $A\_P(a) \circ R\_N(c+a)$ is the mask of the action space. All the masks can be computed in parallel and saved in a table prior to initiating training. Given a transition $c' = c + a$, consulting the table using the tuple $(c, a)$, and the





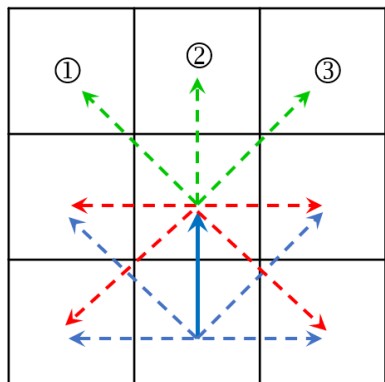 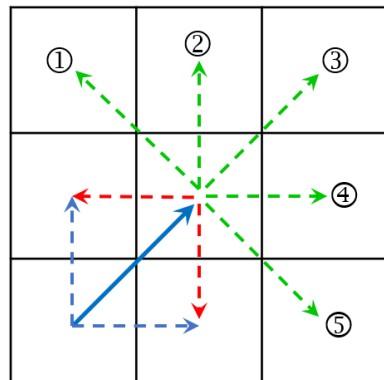

**Figure 10.** Action patterns for up and up-right. The blue solid line denotes the last action, while the red dashed lines represent meaningless next actions. The green dashed lines signify the compressed action space.

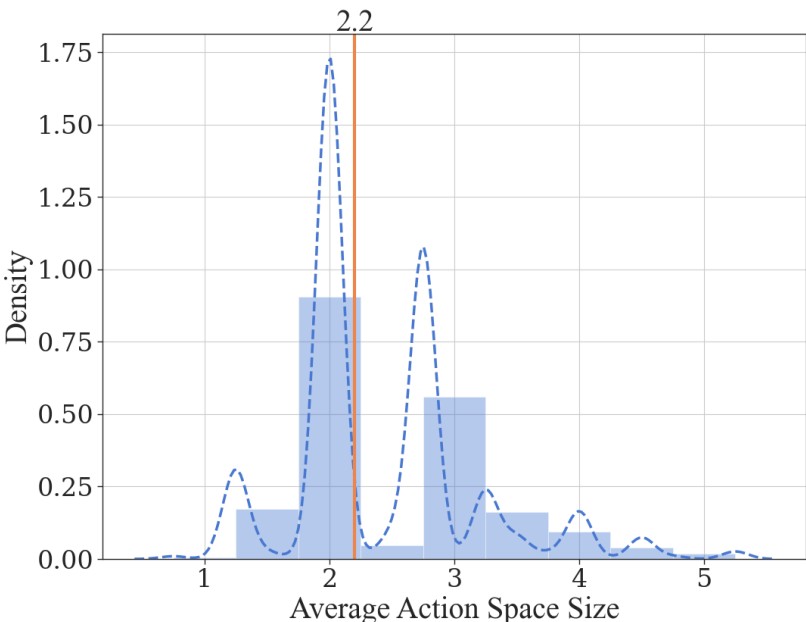

**Figure 11.** The distribution of mean action space size per state. On average, the action space size for each state is less than 3.

action space of $c'$ can be subsequently obtained. A single state corresponds to eight compressed action spaces of varying size, and the distribution of state-averaged action space size is shown in Fig. 11.



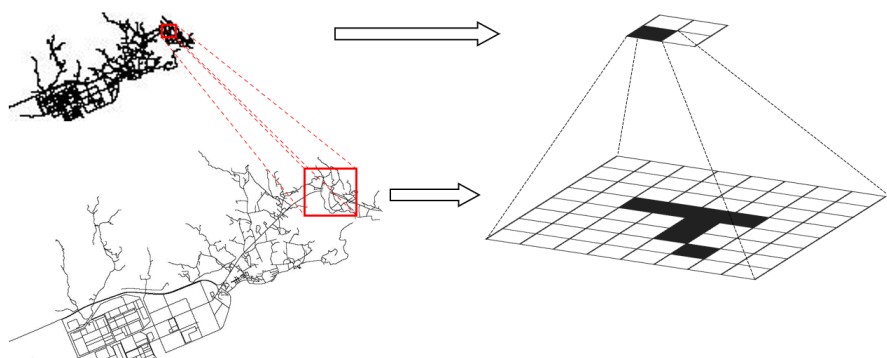

**Figure 12.** The low-resolution image and the high-resolution image. For each cell $c(i,j)$ in the low-resolution image, uniquely corresponds to a rectangular area $rec(i,j)$: $i \times M \leqslant x \leqslant i \times M + M$, $j \times N \leqslant y \leqslant j \times N + N$ in the high-resolution image.

Given start and end points, large-scale path planning often necessitates information pertaining solely to a limited region, as the majority of roads remain untraversed. The challenge lies in discerning which roads are likely to be traversed and which are not. In this study, we introduce the concept of the 'premium region', which encompasses the area the optimal path may traverse, and we proposed a masked state space compression method to determine the 'premium region'.

The base path serves as an auxiliary path, reflecting the optimal path's trend and providing guidance in determining the 'premium region'. We employed an additional low-resolution raster image, obtained through resampling the original raster image. The shortest path on the low-resolution image can be readily computed using a breadth-first search algorithm, with this shortest path functioning as the base path. Expansion of the areas traversed by the base path yields the 'premium region'. The shortest path from any start point on the road network to the nearest shelter can be precomputed, and during training, only the state value within the 'premium' region determined by the base path requires updates. The low-resolution image represents an equally scaled-down projection of the original high-resolution image, as demonstrated in Fig. 12. The cell size of the low-resolution image is set to 128 m×128 m, which is 64 times larger than the cell size of the high-resolution image (16 m×16 m), and the 'shelter range' is 1 cell (128 m). Given the base path $p_l = \{c_0, c_1, ..., c_n\}$, the masked state space is described in Algorithm 2. where $m$ is a 0-1 matrix of size 937×546 serving as a mask, and $\delta$ is the tolerance range. In this study, a suitable tolerance range is $\delta = 6$ (corresponding to an actual distance of 96 m and an actual area of 320 m×320 m). In the example depicted in Fig. 13, based on a red base path derived from the low-resolution image, the blue region in the high-resolution image represents the 'premium region'. By considering only the road information situated in the 'premium region' during the path planning, the state space can be substantially compressed, with a compression ratio below 0.4.

Furthermore, to address the issue of sparse rewards, we proposed the tri-aspect reward mechanism. This mechanism offers a structured approach to reward distribution, categorizing rewards into three distinct aspects: basic rewards, distance rewards, and risk rewards. Basic rewards encourage the agent to reach the goal (shelters) in the fewest steps possible, with goal cells assigned a substantial positive reward (+2000), while other cells receive a negative reward of −1. Distance rewards guide the





---

**Algorithm 2** Computation of the masked state space

**Input:** $p_l, \delta$

**Initialize:** $m[] = \{0\}$

   **for all** cell $c(x, y)$ in $p_l$ **do**

      **for** $i = x \times 8 - \delta$ to $(x + 1) \times 8 + \delta; j = y \times 8 - \delta$ to $(y + 1) \times 8 + \delta$ **do**

        $m[i, j] = 1$

      **end for**

   **end for**

**Output:** $m[] \circ R\_N$

---

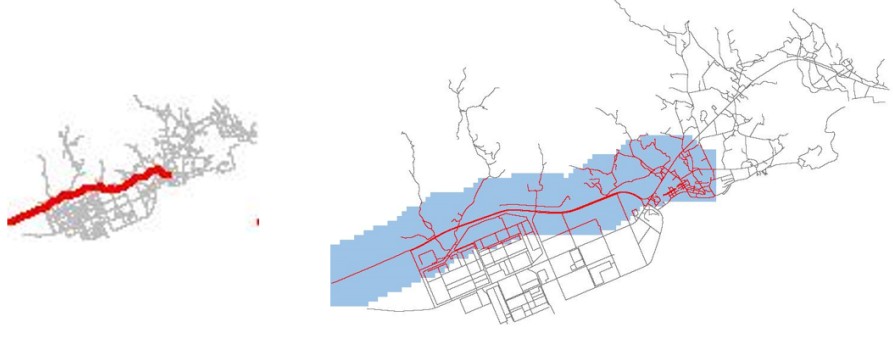

**Figure 13.** An example for masked state space compression.

agent towards the goal, providing a $+2$ reward for moving closer and a $-1$ punishment otherwise. Risk rewards are negative incentives, designed to deter the agent from high-risk cells whenever feasible. There are five risk levels (as depicted in Table 1), and corresponding rewards are illustrated in Table 2. Such a multi-layered reward structure provides the agent with more frequent and meaningful feedback, ensuring a consistent learning trajectory and fostering faster convergence.

**3.2.3 Training process**

Prior to training, the state-action space for each cell within the high-resolution image was computed and stored in a table. Then, the breadth-first search algorithm was applied to the low-resolution image to ascertain the shortest path for each cell to the shelter. During each training episode, the agent commences from a randomly-selected road cell $c_0$. The 'premium region' is





**Table 2.** Risk rewards

| Risk level | Reward |
|:---:|:---:|
| I | 0 |
| II | -4 |
| III | -8 |
| IV | -16 |
| V | -32 |

determined according to the corresponding shortest path in the low-resolution image, and an episode concludes upon the agent
reaching a shelter. At each time step, the input consists of the current cell-centered environmental observation. This observation,
with dimensions $(2r_{ob}+1, 2r_{ob}+1, 4)$, incorporates roads, shelters, risk levels, and minimum Chebyshev distances to shelters
within a $(2r_{ob}+1) \times (2r_{ob}+1)$ rectangular area, where $r_{ob}$ represents the observation range. In this study, the $r_{ob}$ was set to 10
(equivalent to an actual distance of 160 m), reflecting the human field of vision in real-world scenarios. The output comprises
a sequence of length 8, corresponding to the values of 8 actions, and the subsequent action $a_i$ executed by the agent is selected
based on the probability:

$$P(a_i|c) = \epsilon \cdot \frac{1}{|A(\overline{a},c)|} + (1-\epsilon) \cdot \mathcal{F}[Q_\pi(c,a_i), \max(Q_\pi(c,a_j))] \quad (a_i, a_j \in A(\overline{a},c)) \tag{24}$$

where $\mathcal{F}()$ is defined in Algorithm 1; $A(\overline{a},c)$ denotes the action space and it can be obtained by consulting the state-action
space table using the current state $c$ and last performed action $\overline{a}$. $0 \leqslant \epsilon \leqslant 1$ dictates the degree to which selection favors random
exploration over the highest-value action. During the early training stages, a larger $\epsilon$ encourages agents to explore the unknown
environment more extensively. As the model converges, $\epsilon$ should decrease to facilitate agent focus on high-value states and
actions. The agent performs an action to the next cell, subsequently receiving a reward, which serves as an evaluation metric
for the selected action.

Under the DQN framework, the training process is demonstrated in Fig. 14.

## 4    Simulation and Results

In this study, five distinct wind fields were employed to simulate the storm surge for risk assessments, and DQN was utilized
to search for the optimal evacuation path under these scenarios. Three of these scenarios were used as training data, while
the remaining two were used as test data. The study area is a part of coastal suburb of Huizhou with a size of 15 km×9 km.
Historical storm data was utilized in conjunction with the Jelesnianski empirical model to generate a wind field, which was
then provided to the coupled ADCIRC-SWAN model for calculating water levels and simulating inundation extents and depths
within the study area. The DQN was employed to develop evacuation plans, focusing on generating real-time evacuation routes
to predetermined disaster shelters from any given starting position. Extensive experiments were performed, demonstrating the
effectiveness of the proposed method.





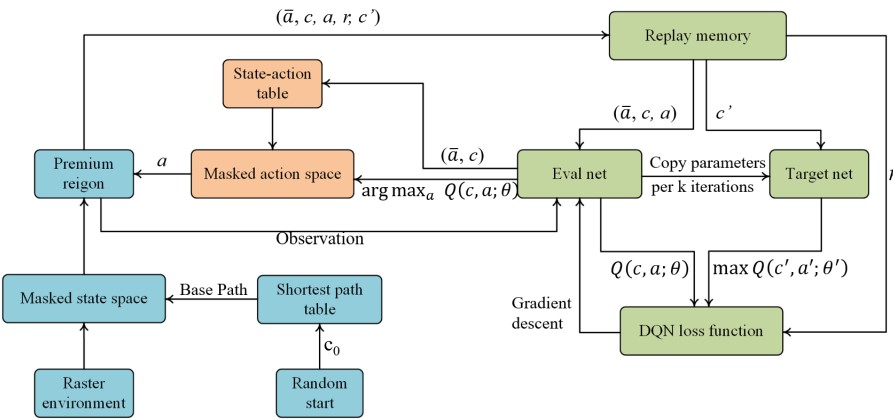

**Figure 14.** The DQN training process with the masked action and state spaces compression, features three components: green sections represent the classical DQN, blue sections correspond to masked state space compression, and orange sections indicate masked action space compression.

## 4.1 Storm surge simulation and risk assessments

To make the hypothetic wind field reasonable, the historical tropical cyclone dataset released by the China Meteorological

Administration (CMA) was used. The dataset contains the location and intensity of tropical cyclones in the northwest Pacific Ocean, recorded every six hours from 1949 to 2018. According to the CMA dataset, the minimum central pressure of tropical cyclones affecting China ranges from 880 hPa to 1000 hPa. Mild storm surges barely penetrate inland, while severe storm surges can inundate the entire area. In this study, five representative typhoon scenarios were defined, with their parameters displayed in Table 3.

**Table 3.** Typhoon scenarios

| Scenario | Minimum central pressure (hPa) | Maximum wind radius (km) | Maximum wind velocity (m/s) |
|---|---|---|---|
| 1 | 910 | 31 | 61 |
| 2 | 920 | 33 | 57 |
| 3 | 930 | 35 | 53 |
| 4 | 940 | 38 | 49 |
| 5 | 950 | 41 | 45 |



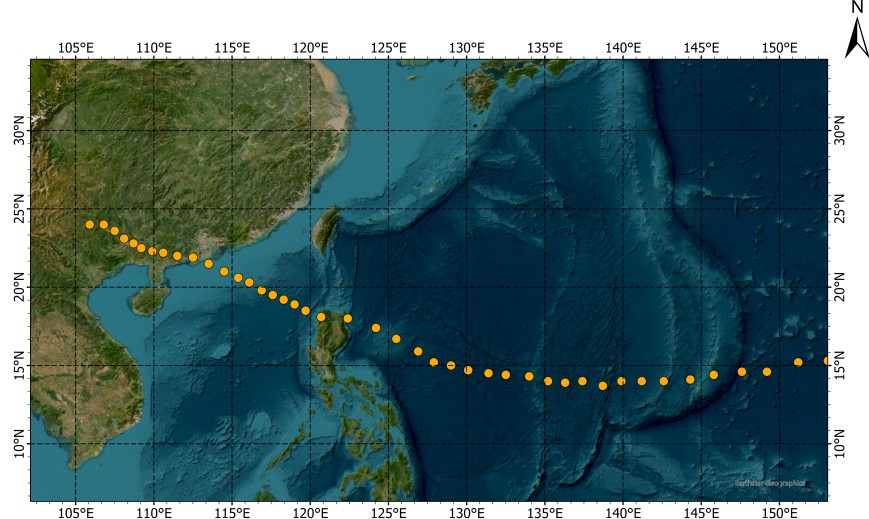

**Figure 15.** The track of Mangkhut.

Additionally, Super Typhoon Mangkhut, which caused the most significant storm surge disaster to the study area on record, had its track chosen as the simulated cyclone's track. The track of Mangkhut is shown in Fig. 15.

Under the wind field generated by the Jelesnianski model, the coupled ADCIRC-SWAN model was run on the datasets to simulate surge water. The inundation depth was calculated by subtracting the DEM from the height of simulated surge water. According to Table 1, the risk assessments were conducted based on the inundation depths. When the inland flooding attains

its maximum extent, the risk assessments for five storm surge scenarios are depicted in Fig. 16.

### 4.2   Model Performance

The goal of this work is to enable real-time planning of the optimal path to the shelter from any given cell based on the surrounding environment. The proposed model achieves high-accuracy evacuation path planning according to the surrounding environment. To evaluate the overall performance of the model, Scenario 1 and 5 were utilized, and 1000 starting cells were

randomly chosen for each test scenario to conduct path planning, respectively. The enumeration method is used to find the true optimal paths under the reward setting for these 2000 locations, which are called the target paths, and the 'optimal' paths output by the model are called the eval paths. In Scenario 1, the target paths cover about 64 % of the road network, with an average length of 4286 m. In Scenario 5, the target paths cover about 61 % of the road network, with an average length of 3758 m.

To evaluate the model performance, the differences between the target paths and the eval paths were measured from two aspects: path similarity and distance to the true destination. Three evaluation metrics were introduced to measure the path similarity including the dynamic time warping (DTW, Müller) rate, the Hausdorff (Huttenlocher et al., 1993) rate, and the




(a) Scenario 1

(b) Scenario 2

(c) Scenario 3

(d) Scenario 4

(e) Scenario 5

**Figure 16.** Risk assessments for five typhoon scenarios.The results were organized and displayed in ArcGis Pro 3.0 software.Risk I areas are considered un-flooded safe areas and are therefore not depicted on the figure. Scenario 1 and 5 were used as test data, while Scenario 2-4 were employed as traingng data. To rasterize the inundation area, take the average inundation depth in a cell as the inundation depth of the cell.





overlap rate. Given a target path $p_{tar}$ of length $l_{tar}$ and an eval path $p_{eval}$ of length $l_{eval}$, the DTW can be discribed as a dynamic programming (DP):

$$\text{Minimize}(D \cdot W) \tag{25}$$

where $D = \{d(p,q)\}$ and $d(p,q)$ is the Chebyshev distance between the cell $p \in p_{tar}$ and cell $q \in p_{eval}$. $W = \{w(i,j)\}$ and $w(i,j)$ is the binary DP variable $i,j \in \mathbb{Z}^+, i \leqslant l_{tar}, j \leqslant l_{eval}$. The above DP is subject to:

$$w(i,j) = \{0,1\} \tag{26}$$

$$w(1,1) = 0 \tag{27}$$

$$w(l_{tar},l_{eval}) = 1 \tag{28}$$

$$w(i,j) = [w(i-1,j) + w(i-1,j-1) + w(i,j-1)] \times w(i,j) \tag{29}$$

The DTW distance is used to measure the average deviated distance of the eval path from the target path that can be obtained by:

$$d_{DTW}(p_{tar},p_{eval}) = \frac{D \cdot W}{n}, \quad n = \sum w(i,j) \quad (w(i,j) \in W) \tag{30}$$

while DTW rate, defined as: $d_{DTW}/l_{tar}$, quantifies the DTW distance per unit length. Consider a path as a set of locations, the Hausdorff distance $d_{Haus}(p_{tar},p_{eval})$, to measure the distance between the two sets of paths, can be computed by:

$$d_{Haus}(p_{tar},p_{eval}) = \max\{\sup\inf d(p,q), \sup\inf d(q,p)\} \quad (p \in p_{tar}, q \in p_{eval}) \tag{31}$$

where $\sup$ is the supremum and $\inf$ is the infimum. The Hausdorff distance measures the maximum deviated distance of the eval path. and the Haudorff rate is: $d_{Haus}/l_{tar}$, which quantifies the Hausdorff distance per unit length. Let $l_{ol}$ denote the

length of the longest overlapping sub-path between the target path and the eval path, the overlap rate is the ratio $r_{ol} = l_{ol}/l_{tar}$. In addition to the similarity, the distance between the destinations of the target path and the eval path can be measured by Chebyshev distance.

By comparing with the target paths, the eval paths can be categorized into optimal paths and deviated paths, where the deviated path differs from the corresponding target path by one or more cells. In Scenario 1, there are 668 optimal paths and

332 deviated paths, while in Scenario 5, there are 620 optimal paths and 380 deviated paths, as shown in Fig. 17.

Focusing on the deviated paths, the DTW rate, the Hausdorff rate, the overlap rate, and the destination distance under Scenario 1 and 5 are illustrated as Fig. 18 and Fig. 19, where one cell is equal to an actual distance of 16 meters. Scenario 1 evaluates the model's performance amidst severe storm surges and complex inundation environments, while Scenario 5 assesses the model's performance in environments with minimal flooding. Among the 2000 eval paths tested in Scenario 1

and 5, 668 and 620 paths are optimal, respectively. For the remaining deviated paths, on average, the DTW distance is 76.8 m per path and 0.04 m per meter in Scenario 1, and 65.6 m per path and 0.03 m per meter in Scenario 5. The Hausdorff distance is 168 m per path and 0.07 m per meter in Scenario 1, and 252.8 m per path and 0.08 m per meter in Scenario 5,



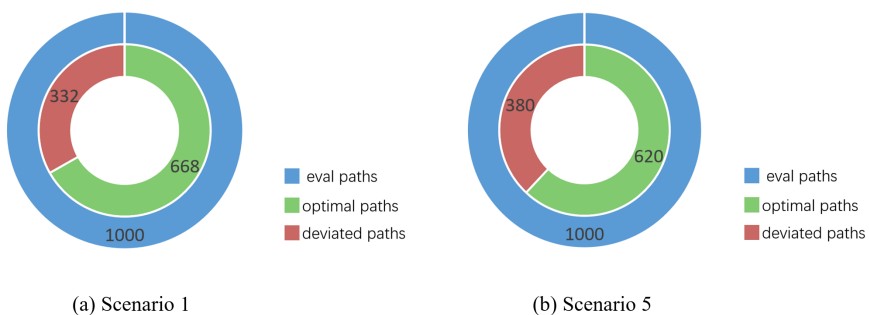

(a) Scenario 1          (b) Scenario 5

**Figure 17.** The number of optimal paths and deviated paths for Scenario 1 and 5.

signifying that the eval paths exhibit a remarkable morphological similarity to the target paths. The mean overlap rate is 0.74 and 0.71, respectively, signifying that a substantial portion of the eval paths coincides with the target paths. Furthermore, the mean destination distance is 78.4 m and 107.2 m, ensuring that evacuees reach a location very close to the most suitable shelter. Notably, the mean lengths of the deviated paths (5584 m and 4832 m), are longer than the mean lengths of eval paths (4826 m and 3758 m), revealing the model's inadequacies in effectively planning long path. Overal, these deviations are negligible for the study area of 135 km². The experimental results demonstrate that the proposed method exhibits strong performance in generality, providing emergency evacuation path planning for the entire study area.

## 5 Conclusions

This study presents a comprehensive approach to emergency evacuation planning in the Daya Bay Petrochemical Industrial Zone. By coupling a risk assessment of storm surges with a road network, a raster environment that reflects real-world scenarios has been constructed. The DQN model was employed to develop a real-time evacuation plan, providing efficient and effective guidance for individuals during storm surge events. To enhance the adaptability of the DQN model for rasterized road network, masked state space, masked action space, and tri-aspect reward mechanism were proposed, significantly enhancing the model's convergence. The coupled ADCIRC-SWAN model and Jelesnianski method were used to create the simulation environment of storm surges under different typhoon scenarios. Additionally, potential safe shelters were identified for the study area to provide more evacuation options.

Two distinct storm surge scenarios with the minimum central pressures of 910 hPa and 950 hPa were used as the test environments, and path plan for 1000 randomly selected starting cells were conducted in each scenario. By comparing the eval paths with the target paths, for both scenarios, over 60 % of the eval paths were optimal, while the remaining 40 % exhibited only minor deviations from the optimal paths, with an average difference of merely 4 centimeters per meter and an average overlap rate of exceeding 70 %. Moreover, the average destination distance between the deviated paths and the optimal paths



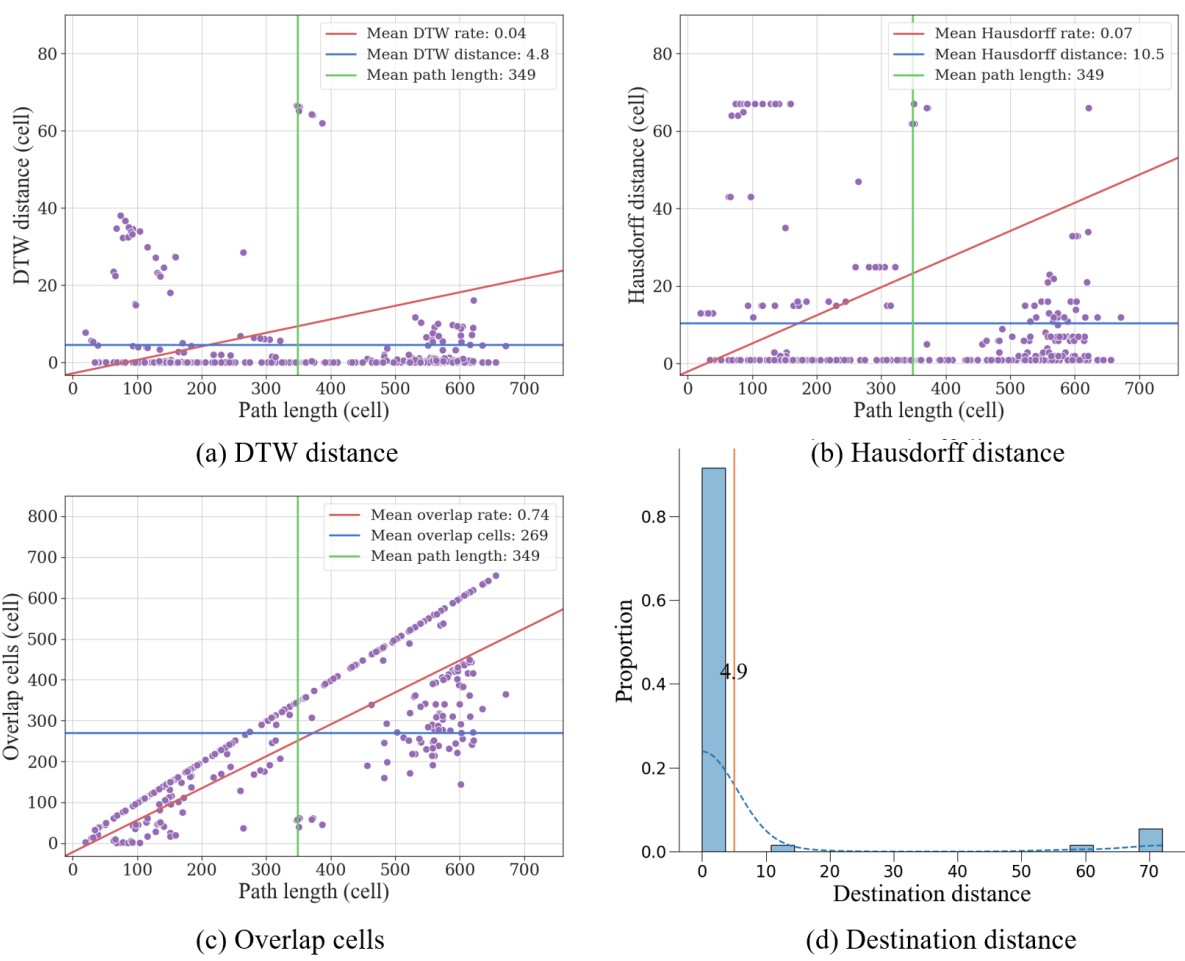

**Figure 18.** Model performance in Scenario 1.





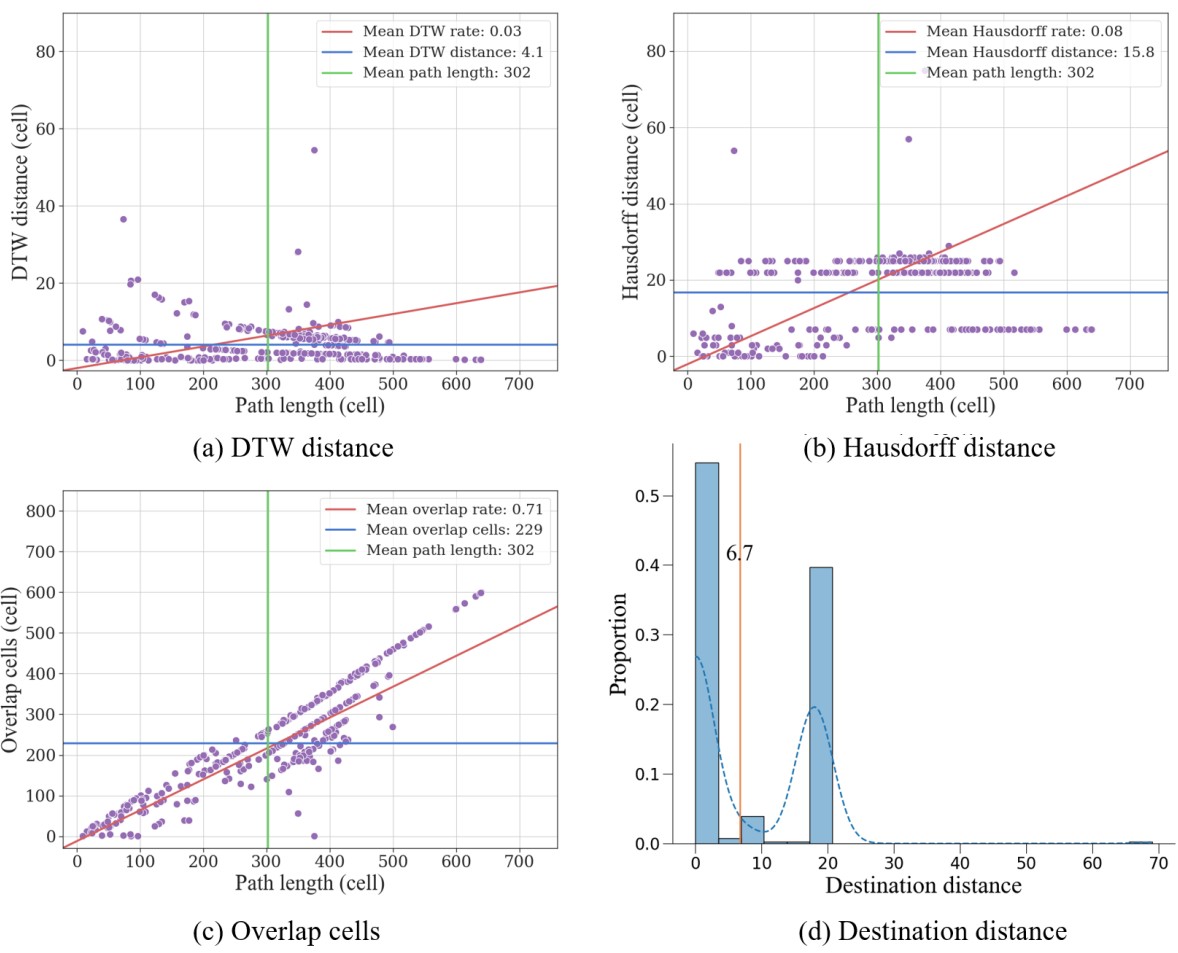

(a) DTW distance

(b) Hausdorff distance

(c) Overlap cells

(d) Destination distance

**Figure 19.** Model performance in Scenario 5.




was approximately 100 m. Overall, the proposed method proved highly effective in planning optimal evacuation routes with a
minimal deviation that can be of great assistance for the evacuee during a real-world storm surge.

Based on the results and findings presented in this study, the proposed method showed effectiveness in enhancing real-time
emergency evacuation plans and demonstrated the potential of employing advanced modeling techniques to improve emergency
response and preparedness in vulnerable areas. However, there is still room for improvement and future work can be done to
further optimize the evacuation plan. One potential direction involves replacing the raster environment with a topological envi-
ronment and utilizing graph neural network techniques. Due to the limitations of the raster environment, the proposed method
trains slowly and is difficult to apply on a larger scale area (such as a city area). The reason for adopting the raster environment
in this paper is that it is simple to construct and can directly correspond with the flooding extents. Another possible avenue for
future research is to incorporate more advanced machine learning algorithms or data-driven models, as there is potential for
further improvement with currently 60 % of eval paths being optimal. Additionally, more environmental information could be
utilized to meet more complex demands, such as incorporating population restrictions for shelters, identifying different types
of roads' passing conditions and costs, and so on. This study is a promising start in developing real-time emergency evacuation
plans, and we eagerly anticipate future advancements in this field.

*Author contributions.* Yan Li and Wenjuan Li designed the research and optimized the overall structure of this paper. Chuanfeng Liu and Si
Wang completed most of the main work, including the programming, debugging of parameters, and final drafting of the article. Lin Mu and
Darong Liu contributed some important algorithm ideas and completed the work of the comparison part. Hao Qin and Kai Zhou provided the
original algorithm ideas and framework for this study and provides valuable suggestions for program optimization and parameter adjustment.

*Competing interests.* The authors declare that they have no conflict of interest.

*Financial support.* This work was supported by Shenzhen Science and Technology Program (Grant No.KCXFZ20211020164015024).



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
