# Peer review of "Effective Storm Surge Risk Assessment and Deep Reinforcement Learning Based Evacuation Planning: A Case Study of Daya Bay Petrochemical Industrial Zone"

_EGUsphere, 2023_

## Author Response (AR1)

**Dear Editors and Reviewers:**

Thank you for your letter and for the reviewer' comments concerning our manuscript entitled "Effective Storm Surge Risk Assessment and Deep Reinforcement Learning Based Evacuation Planning: A Case Study of Daya Bay Petrochemical Industrial Zone" (ID: egusphere-2023-2280). Those comments are all valuable and very helpful for revising and improving our paper, as well as the important guiding significance to our research. We have carefully revised our manuscript which we hope meet with approval. We are uploading our point-by-point response to the comments, an updated manuscript, and a marked-up version of our revised manuscript. The main corrections in the paper and the response to the reviewers' comments are as following:

**Review 1:**

This study proposes a method for Storm Surge Evacuation Planning using a coupled Deep Q-Network (DQN) model, ADCIRC, and SWAN models. However, the research appears more like a report rather than a scientific study, as many parts are not clearly explained. Therefore, a major revision is required. I would like to suggest the following improvements for the manuscript:

- 1. Introduction: authors discuss the differences between traditional methods and the DQN method. However, it is difficult for me to understand the specific benefits of using Deep Reinforcement Learning (DRL) to improve evacuation planning. It would be helpful to provide more information on how DRL can enhance the evacuation process, highlighting the innovation of this work.
- 2. Figures: There are too many figures, and most of them could benefit from more detailed information, and enhance the layout of the figures. For example, Fig. 1 and Fig. 4 could be merged into a single figure. Additionally, adding more descriptive captions to the figures would be beneficial.
- 3. Validation: It is unclear where the authors validate the ADCIRC and SWAN models using real historical disaster events. Usually there would be some plot for real tide gauges. Please provide information on the validation process and the results obtained.
- 4. Methodology: Page 23, Lines 365-375: This section seems to belong to the methodology rather than the results. Please consider moving it to the appropriate section.
- 5. It would be helpful to explain the relationship between the Markov Decision Process and the DQN algorithm in the text. This would ensure a smoother transition and better understanding for the readers.
- 6. Results: The results regarding the optimal evacuation paths are presented in a simple and unclear manner. Please provide additional information and

- clarification to improve the quality of the results section. It is important to avoid giving the impression of being careless or sloppy.
- 7. Title: Please use the full name "Deep Reinforcement Learning (DRL)" in the manuscript title to provide a more accurate representation of the study.

I hope these suggestions help in improving the manuscript. Please ensure there are no grammatical errors in the revised version.

**Response to review 1:**

We would like to express our sincere gratitude for your professional advice. These comments help to improve rigor of our study. We have carefully considered your comments and have thoroughly revised the manuscript according to your suggestions. This study is a deep dive into the realm of interdisciplinary research, incorporating theories and methods from various disciplines. We acknowledge that while such interdisciplinary integration offers a broad perspective and enriches insights, facing challenges in articulating it systematically is inevitable. We regret for any parts where we have not clearly explained. We have further refined our article to ensure its presentation is cogent. We would like to show the details as follows:

- 1. Conventional evacuation route planning is limited by its oversimplification of complex environments and inadequate consideration of victims' needs. The unpredictability of storm surges and incomplete regional information further complicate the maintenance of effective plans. The benefits of using deep reinforcement learning to address path planning issues lies in its ability to automatically select safe roads and plan appropriate evacuation routes for affected individuals based on their limited surrounding environment. This approach can adapt to dynamically changing disaster environments, providing guidance for individuals during storm surge disasters. We have revised the Introduction section to enhance the presentation of the benefits of using DRL (refer to Pages 2 & 3, Lines 49—62), and we listed the innovative aspects of our study in the Introduction (refer to Page 3, Lines 75—86).
- Considering your feedback on the excessive number of figures and the need for more detailed information, we deleted and modified some figures, where Figure
   and Figure 3 were merged; Figure 5, Figure 6 and Figure 11 were deleted;

- Figure 12 and Figure 13 were merged; Figure 16, Figure 18, Figure 19 were modified. We believe these improvements will more effectively convey the information and enhance the overall quality of the article.
- 3. We had actually used water levels observation data from hydrological stations to validate the ADCIRC+SWAN model, and the real historical typhoon events (2305, 2311, 2314) were used to validate water levels. The reason why this part of the verification was not included in the main text is primarily because the focus of this study leans more towards evacuation path planning rather than storm surge simulation. The verification of water levels is not as crucial. We have included the validation results for the ADCIRC+SWAN model in the Appendix A.
- 4. We have thoroughly revised the content of the Results section as suggested. The content you referred to has introduced the evaluation metrics of the DQN model, and placed it within the Results section is appropriate. However, recognizing that this part was overly lengthy and complex, we have made amendments to condense and streamline the content (refer to Page 21, Lines 333—339).
- 5. The Markov Decision Process is the mathematical foundation of the DQN algorithm. To leverage the advantages of deep reinforcement learning, we transform the path planning problem into a Markov Decision Process problem, and then employed the DQN model to address it. We apologize for not making this part clear in the article and we have enhanced the explanation of the relationship between the Markov Decision Process and the DQN algorithm in the evacuation route planning subsection of Method Section. We have provided a detailed explanation between the Markov Decision Process and the DQN algorithm in Section 3.3 (refer to Page 11 &12, Lines 180—219).
- We have completely revised the contents on evacuation route planning in Section
  and modified the relevant figures. The revised section is now more concise,
  but the presentation is clearer (refer to Page 19-22).
- 7. We have integrated the full name "Deep Reinforcement Learning" into the manuscript's title to more accurately reflect the content of the study ,and the revised title name is "Effective Storm Surge Risk Assessment and Deep

Reinforcement Learning Based Evacuation Planning: A Case Study of Daya Bay Petrochemical Industrial Zone"

Thank you again for your suggestions, these will help in improving the manuscript, and we ensured there are no grammatical errors in the revised manuscript.

**Review 2:**

The study addresses an important problem in coastal areas prone to storm surges, such as the Daya Bay Petrochemical Industrial Zone. The coupling of risk assessment models with a road network and the utilization of DRL for path planning is well-motivated and logical. The authors have proposed several methods to address convergence challenges in their DRL model, such as masked state space, masked action space, and a tri-aspect reward mechanism which effectively enhance the model's convergence capabilities. The analysis conducted using the coupled ADCIRC+SWAN model for simulating storm surges and evaluating risk assessments provides insights into inundation depths and extents during various typhoon scenarios. Additionally, the evaluation of optimal evacuation routes using DRL demonstrates promising results in terms of path similarity and distance to true destinations. However, I would like to suggest some revisions that would strengthen the manuscript.

- 1. In abstract, I suggest to add the study aim after the research gap is identified (However, in practical...), claiming the research objective that is tried to solve.
- 2. In introduction, why ADCIRC+SWAN model employed for the study? The advantages of this model over other models need to be stated.
- 3. In introduction, the state-of-the-art of DRL implications in emergency evacuation should be provided. If there are other studies employing DRL in emergency evacuation, the literature review should be conducted and claim the difference and originalities between the current study and previous studies. If this is the first application, please clearly claim in the introduction.
- 4. Please clarify the specific objective of the study and its significance in addressing research gaps in storm surge evacuation in the introduction.
- 5. Please clearly state the originalities and limitations of the current research in the introduction.
- 6. Literature review is not very updated. Please update the literature review until recent studies in 2023 and 2024.
- 7. Line 122-123, 11 major tidal components were included in the study, and please explain the reason of choosing these components.

- 8. In part 4.1, please add ADCIRC+SWAN model calibration with historical data, enhancing the credibility of the flood simulation and risk assessment. After the model calibration, the typhoon scenarios and corresponding risk assessment could be further studied.
- 9. Please include discussion of uncertainty or parameter sensitivity analysis associated with numerical models for storm surge simulations.
- 10. In conclusion, please add a section of concise summary of keys findings, and discuss how these findings can be applied in real-world decision-making process.

**Response to review 2:**

We deeply appreciate your pointing out the areas where our manuscript requires further substantiation, and your constructive suggestions for improvements. Your comments are invaluable in improving our manuscript and advancing research in the field. We have addressed each of the concerns raised and have worked diligently to revise our manuscript accordingly. Taking the Daya Bay Petrochemical Industrial Zone as an example, this study integrates risk assessment with the road network and employs deep reinforcement learning algorithm for evacuation path planning. It can provide effective guidance for affected individuals based on the limited surrounding environment. This study focuses on the evacuation path planning for storm surges, and thus the description of the storm surge simulation part is rather brief, rendering the article somewhat sketchy and unclear. We have revised the manuscript according to the suggestions provided, as follows:

- 1. As suggested, we claimed the research objective that is tried to solve in the abstract. This addition clarifies our research objectives and enhances the abstract's coherence (refer to Page 3, Lines 66–86).
- 2. In the Introduction section, we have presented the features of the ADCIRC and SWAN models (refer to Page 2, Lines 37—40), and quoted relevant literature to show that the ADCIRC+SWAN model has been widely applied worldwide and has achieved good performance (refer to Page 2, Lines 41—42). The ADCIRC+SWAN model is currently a more mature model used for storm surge simulations, incorporating both wave and current factors, with more accurate simulations of storm surges. Additionally, the two models share a same grid for

- synchronous coupling, simplifying usage.
- 3. We have added some related works on DRL based evacuation route planning (refer to Page 3, Lines 58 61). However, this study represents the first application of to combine the risk assessment and DRL based evacuation route planning within large-scale raster environments. We have summarized the differences between our research and others, and clarified the innovations in main contributions in the Introduction (refer to Page 3, Lines 66–86).
- 4. In the introduction, we have delineated the specific objectives and highlight the significance of our study in addressing research gaps in storm surge evacuation. We added a new content of main contributions to summarize our works, aiming to offer readers a concise overview of the unique contributions of our research (refer to Page 3, Lines 66–86).
- 5. We have updated the Introduction section to explicitly state the original contributions and limitations of current research. In the tail of the Introduction we added a summary of the limitations of the current research and main contributions of this work. Additionally, we summarized the value and significance of our study in the conclusion section for its application in the real world (refer to Page 3, Lines 66–86; Page 22 & 23, Lines 357–372).
- 6. We have updated the literature review to include the latest advancements and discussions in the field in Introduction section.
- 7. In the South China Sea, the predominant tidal components are four diurnal and four semidiurnal components. We have also included three additional components, which actually have a minor impact on the simulation results. Given that our research concentrates on storm surge risk assessment and evacuation path planning, the precision requirements for storm surge simulation are less stringent. Consequently, we employed these 11 tidal components. In the Section 3.1, we briefly explained why we chose these tidal components (refer to Page 7, Lines 134—136).
- 8. We had actually used water levels observation data from hydrological stations to validate the ADCIRC+SWAN model, and the real historical typhoon events

(2305, 2311, 2314) were used to validate water levels. The reason why this part

of the verification was not included in the main text is primarily because the focus

of this study leans more towards evacuation path planning rather than storm surge

simulation. The verification of water levels is not as crucial. We have included

the validation results for the ADCIRC+SWAN model in the Appendix A.

9. This study focuses on the risk assessment and the evacuation path planning. In

setting up of the coupled model, we adopted several existing parameterization

schemes and obtained marginally satisfactory simulation results. Given that our

focus is not on the numerical simulation of storm surges, we did not perform a

parameter sensitivity analysis of the numerical model.

10. We have enhanced the conclusion section by providing a concise summary of the

key findings and discussing the applicability of this work in practical decisionmaking processes (refer to Page 23, Lines 362 – 375), as well as the limitations

of our work and the directions for future research (refer to Page 23, Lines 375—

382).

We hope these responses address the concerns raised. We are grateful for the

opportunity to enhance our work based on the valuable suggestions received and look

forward to the potential contribution of our study to the literature in this area of research.

Corresponding Author:

Name: Chuanfeng Liu

Affiliation: College of Oceanic and Atmospheric Sciences, Ocean University of China.

E-mail address is rzlcf1998@163.com

Yours sincerely,

Chuanfeng Liu, Yan Li, Wenjuan Li, Hao Qin, Lin Mu, Si Wang, Darong Liu, and Kai

Zhou

---

## Author Response (AR2)

Dear Editors and Reviewers,

Thank you for your insightful comments and constructive feedback on our manuscript. We have carefully considered all your suggestions and have revised the manuscript accordingly. We believe these changes have significantly improved the clarity, rigor, and overall quality of our paper.

Please find our point-by-point responses to your comments below.

**Comment 1:** Please explain how you determine the transition probabilities. Did authors use any real data to train the transition probabilities value?

**Response to Comment 1:**

Thank you for this important question. We have added a detailed clarification in Section 3.3 of the revised manuscript.

In our study, the transition probability is not trained from data but is determined by the nature of the environment model. It is important to clarify the nature of the transition probability in our model. In a general Markov Decision Process (MDP), the state transition function can be stochastic, representing inherent uncertainty in an environment's dynamics. In contrast, **the raster-based environment in this study is deterministic**. Specifically, any action taken from a given state (a grid cell) deterministically leads to a single, known subsequent state (the adjacent cell). This means the transition probability is 1 for the resulting state and 0 for all others. As the environment's transition model is perfectly known and deterministic, **it does not need to be learned or estimated from real data.** The agent's learning challenge is therefore focused not on modeling the environment's dynamics, but on discovering the optimal policy within these known dynamics.

**Comment 2:** Please find the reference to the reward function structure. Where to the values in Table 4 come from? Please give a detailed explanations.

**Response to Comment 2:**

Thank you for the valuable suggestion. We have significantly expanded the explanation of our reward function design in Section 3.4 and included relevant citations.

The reward function detailed in Table 4 was designed heuristically based on the principle of **reward shaping** (Ng et al., 1999; Ibrahim et al., 2024), a common approach for creating dense

and informative signals to accelerate learning. The specific values were calibrated empirically against the environment's scale. Our analysis indicated that a medium-length evacuation route comprises approximately 100 steps, which established the step reward as a baseline unit. The destination reward (+100) was thus set to counteract the cumulative penalty of a medium-length path, while risk penalties were scaled significantly higher to prioritize safety over minor efficiency gains. This empirically-grounded calibration proved highly effective in guiding the agent toward optimal and safe evacuation routes.

**Comment 3:** Please add a citation to Figure 7. It is a very generous RL model figure, so you need to cite the references.

**Response to Comment 3:**

Thank you for pointing this out. Figure 7 indeed illustrates the standard reinforcement learning framework. We have revised the caption for Figure 7 to properly credit the foundational work in the field. The caption now begins with: "The DRL model, **adapted from Sutton and Barto**."

**Comment 4:** Please explain how your deep learning network is trained. Please specify the input data and output results. The results should validate why deep learning is imperative for this research.

**Response to Comment 4:**

Thank you for this insightful comment, which has prompted us to significantly improve the structure and clarity of our methodology section. We have reorganized Section 3.4 to first explain why a deep learning approach is imperative, and then detail how the network is trained.

1. Justification for Deep Learning: We have added a new paragraph at the beginning of Section 3.4 to address this. Deep reinforcement learning is imperative for this research for several key reasons.

Firstly, the problem is characterized by a **high-dimensional state space**. The agent's state is not a simple coordinate but a rich, image-like observation, rendering traditional tabular methods computationally infeasible. More importantly, a deep neural network provides powerful **generalization**, learning abstract environmental features—such as intersections or flooded zones—rather than memorizing individual states. This allows the agent to make intelligent decisions in novel, previously unseen situations. Lastly, its end-to-end learning capability allows for policy optimization directly from raw environmental data, bypassing manual feature engineering.

2. Input, Output, and Training Process: We have also added a new, consolidated paragraph that clearly defines the training mechanics.

Input: At each step, the network takes the agent's local observation as input—a multi-channel tensor of size (2rob+1,2rob+1,4) containing information on roads, shelters, risks, and distances. Output: The output is a vector of 8 Q-values, predicting the expected return for each possible action. Training: The network's weights are optimized by minimizing the Mean Squared Error between the predicted Q-values and target Q-values. To ensure stable training, this process utilizes a separate target network and samples experiences from a replay memory to break temporal correlations.

We believe this new structure better addresses your questions and improves the overall readability of the manuscript.

Once again, we sincerely thank you for your time and for providing such valuable feedback to help us improve our work.

Sincerely,

Chuanfeng Liu, Yan Li, Hao Qin, Wenjuan Li, Lin Mu, Si Wang, Darong Liu, and Kai Zhou